# A Bionic Venus Flytrap Soft Microrobot Driven by Multiphysics for Intelligent Transportation

**DOI:** 10.3390/biomimetics8050429

**Published:** 2023-09-17

**Authors:** Xiaowen Wang, Yingnan Gao, Xiaoyang Ma, Weiqiang Li, Wenguang Yang

**Affiliations:** 1School of Electromechanical and Automotive Engineering, Yantai University, Yantai 264005, China; wxw2435461356@163.com (X.W.); m17865570339@163.com (Y.G.); 18854376225@163.com (X.M.); 2School of Accounting, Shandong Youth University of Political Science, Jinan 250000, China

**Keywords:** biologically inspired soft robot, multi-stimulus-responsive, reversible deformation

## Abstract

With the continuous integration of material science and bionic technology, as well as increasing requirements for the operation of robots in complex environments, researchers continue to develop bionic intelligent microrobots, the development of which will cause a great revolution in daily life and productivity. In this study, we propose a bionic flower based on the PNIPAM–PEGDA bilayer structure. PNIPAM is temperature-responsive and solvent-responsive, thus acting as an active layer, while PEGDA does not change significantly in response to a change in temperature and solvent, thus acting as a rigid layer. The bilayer flower is closed in cold water and gradually opens under laser illumination. In addition, the flower gradually opens after injecting ethanol into the water. When the volume of ethanol exceeds the volume of water, the flower opens completely. In addition, we propose a bionic Venus flytrap soft microrobot with a bilayer structure. The robot is temperature-responsive and can reversibly transform from a 2D sheet to a 3D tubular structure. It is normally in a closed state in both cold (T < 32 °C) and hot water (T > 32 °C), and can be used to load and transport objects to the target position (magnetic field strength < 1 T).

## 1. Introduction

Intelligent robot technology has made great progress, and more and more robot technologies have leapfrogged traditional industrial applications and entered many fields, such as healthcare [1,2,3,4,5,6,7], education and entertainment, and intelligent transportation [8,9,10]. The development of flexible cordless robots has become crucial, as the high requirements for robots to operate in complex environments increase. Inspired by nature, researchers are working on smart materials with reversible deformation properties, such as stimulus-responsive hydrogels [11,12,13,14,15] and liquid crystal elastomers [16,17,18,19,20,21,22,23]. Among these several smart soft materials, hydrogels stand out because of their wonderful biocompatibility and deformation reversibility and have been being used by more and more researchers for the manufacture of soft robots [24,25,26,27,28,29,30,31].

Poly (N-isopropyl acrylamide) (PNIPAM) hydrogel is a type of temperature-sensitive hydrogel [14,32,33,34]. The volume of PNIPAM hydrogel can mutate near its critical dissolution temperature (LCST = 32 °C). When the temperature is below the LCST (<32 °C), its volume expands rapidly, and when the temperature is above the LCST (>32 °C), it shrinks rapidly. The PNIPAM hydrogel prepolymer is cured to 2D sheets with a non-uniform expansion field inside by ultraviolet (UV) curing technology. These 2D sheets can deform into a 3D structure upon specific external stimulations. To maintain the 3D structure, even in the absence of stimulation, we cure a polyethylene glycol diacrylate (PEGDA) hydrogel layer on the PNIPAM layer. PEGDA hydrogel does not exhibit significant changes in its swelling ratio upon a change in temperature. Hence, for this bilayer structure, the PNIPAM layer acts as the active layer, while the PEGDA layer acts as the rigid layer.

Additionally, the swelling ratio of PNIPAM hydrogel can respond not only to a change in temperature, but also to a change in the solvent composition of solutions (Figure 1). The volume of PNIPAM reaches its maximum only in pure water or pure ethanol. In a mixed solution of ethanol and water, PNIPAM shrinks. In contrast, the swelling ratio of PEGDA hydrogel does not change significantly regardless of whether it is in either solution. In this research, we fabricated PNIPAM sheets with microchannels and took advantage of the temperature response characteristics of PNIPAM to achieve self-folding. Subsequently, an additional PEGDA layer was cured on the surface of the PNIPAM sheet to obtain the PNIPAM–PEGDA bilayer structure. We fabricated bionic flowers and bionic soft robot Venus flytraps based on this bilayer structure, which can be used to wrap and transport objects under the effect of an external magnetic field. This study provides a reference for the programmable deformation of smart soft materials, which has great prospects in intelligent transportation, bionics, and biomedicine fields.

## 2. Materials and Methods

### 2.1. Materials

PNIPAM prepolymer is made up of NIPAM monomer (>98%), BIS (>99%), and TPO. Among these, NIPAM monomer (>98%) and BIS (>99%) were purchased from Aladdin Biochemical Technology Co. (Shanghai, China), and TPO was purchased from Sigma-Aldrich (Burlington, MA, USA). PEGDA prepolymer is is made up of PEGDA monomer and TPO. PEGDA monomer was purchased from Sigma-Aldrich (United States). The photothermal conversion capability and magnetic response characteristic of the robot are provided by CNTs and Fe_3_O_4_, respectively. The CNTs used are multi-walled carbon nanotubes (MWCNTs), which were purchased from Tianfeng Graphene Technology Co., Ltd. (Suzhou, China) and their diameter was 100 nm. Fe_3_O_4_ was purchased from Macklin Biochemical Co., Ltd. (Shanghai, China). In addition, experimental consumables such as ethanol, PI film, and FEP film were purchased from Taizhou Taifiilong Products Co., Ltd., (Taizhou, China).

### 2.2. Preparation of PEGDA Prepolymer

We prepared the PEGDA prepolymer solution with a monomer content of 20%. A total of 40 mg of PEGDA monomer and 1 mg of TPO were added into a beaker, and 47.7 µL of ethanol and 111.2 µL of DI water were injected into the beaker. This beaker was placed on the magnetic stirring platform for about 30 min to obtain uniform PEGDA prepolymer.

### 2.3. Preparation of PNIPAM Prepolymer

A total of 96 mg of NIPAM monomer, 24 mg of TPO, and 26 mg of BIS were added into a beaker, and then 400 µL of ethanol and 1400 µL of DI water were injected into the beaker. This beaker was placed on the magnetic stirring platform for about 30 min to obtain uniform PNIPAM prepolymer.

### 2.4. Fabrication of Bilayer Structure

The PNIPAM–PEGDA bilayer structure was fabricated via a secondary UV-curing technology. A PNIPAM layer with a thickness of 400 µm was first prepared and then the excess prepolymer on the surface of it was wiped off. A small amount of PEGDA prepolymer was added onto the PNIPAM layer, and the thickness of PI layer was increased to proceed with the second curing. Ultimately, the bilayer structure was obtained.

## 3. Results

### 3.1. Fabrication of Light-Driven PNIPAM Sheet

Figure 2a,b depicts the UV curing process of the PNIPAM hydrogel sheet that can undergo programmable deformation employed in this study. PNIPAM prepolymer is composed of NIPAM monomer, photo-initiator diphenyl (2,4,6-trimethylbenzoyl) phosphonium oxide (TPO), and photo-crosslinker methylacrylamide (BIS). UV-curing technology based on a digital micromirror device (DMD) has the advantages of high precision and high efficiency, and can be used to cure most photosensitive resins. The device consisted of a UV light source, DMD, and a light curing platform (Figure 2b(1)). The FEP film was attached to two glass slides to facilitate the peeling of the hydrogel sheet. The thickness of the hydrogel sheet was determined by the thickness of the polyimide (PI) spacer, as illustrated in Figure 2b(2). Hydrogel structures with a specific shape and anisotropy can be fabricated by controlling the shape and gray level of the mask.

To make the hydrogel sheet capable of photothermal conversion and magnetic response characteristics, we impregnated it in CNT/Fe_3_O_4_ solution (Figure 2c). Firstly, the hydrogel sheet was placed in hot deionized (DI) water above 32 °C and water was lost from the micropores inside the hydrogel, resulting in a decrease in the volume of the hydrogel sheet. Subsequently, the hydrogel sheet was placed into cold DI water dispersed with CNTs and Fe_3_O_4_. Water reentered the micropores inside the hydrogel, along with the CNTs and Fe_3_O_4_. Repeat the above procedure to obtain the PNIPAM sheet with photothermal characteristics (Figure 2d).

### 3.2. Deformation Properties of PNIPAM Sheet

When one side of the PNIPAM sheet was illuminated by light, the CNTs on the surface rapidly absorbed the light energy and converted it into heat. The illuminated side lost water and decreased in volume, while the volume of the backlit side remained unchanged, bending the hydrogel toward the light source. After turning off the light, the heat rapidly transferred to the surrounding water and the temperature dropped, causing the illuminated side of temperature-responsive hydrogel to reabsorb water, and the deformation recovered. During the experiments, we found that the ethanol content of the hydrogel affected its deformability. We prepared prepolymers containing 5%, 20%, 30%, 40%, and 45% ethanol, and prepared hydrogel sheets with the same shape and size, and a thickness of 400 µm. As illuminated in Figure 3a(1),(2), when the ethanol content was below 30%, the bending angle increased proportionally with the ethanol content. When the ethanol content was 30%, the bending angle reached the maximum. However, when the ethanol content exceeded 30%, the bending angle decreased with the increase in ethanol content. Therefore, the ethanol content of the PNIPAM prepolymer solution was set to 30% in this research. In addition, the thickness of sheet affected its response time, bending angle, and recovery time after illumination. We prepared hydrogel sheets with thicknesses of 200 µm, 300 µm, 400 µm, 500 µm, and 600 µm. The upper edge of each sheet was clamped with tweezers and placed in cold water (about 22 °C). The sheets sagged naturally under the effect of gravity. When one side of each sheet was illuminated with light, the hydrogel sheets with various thicknesses were deformed to a certain extent. As shown in Figure 3b(1), when the sheet was too thin (<200 µm) it floated because gravity could not overcome the buoyancy force. When the thickness of the hydrogel sheet was below 400 µm, the bending angle under illumination increased proportionally with the thickness increase. When the thickness exceeded 400 µm, the bending angle gradually decreased. In addition, with the increase in the sheets’ thickness, the response speed gradually became faster and the recovery speed gradually slowed down (Figure 3b(2)). The above phenomenon can be explained by the following facts. As the thickness of the sheet increases, more water is expelled from the illuminated side in a certain period of time, so the deformation speed is accelerated. In addition, when the thickness of the sheet is less than 400 µm, the illuminated side loses less water and therefore the deformation is small. With the increase in the thickness of the sheet, the water loss from the illuminated side increases, and the bending angle also increases. Therefore, after removing the light source, the illuminated side needs a longer time for water absorption, resulting in the recovery time increasing with the increase in thickness. As shown in Figure 3b(3), the PNIPAM sheet with a thickness of 400 µm is the most appropriate choice for our experiments.

### 3.3. Research on the Swelling Ratio of PNIPAM–PEGDA Bilayer Structure

The PNIPAM hydrogel is sensitive to temperature and can respond quickly to temperature changes. There is a certain proportion of hydrophobic and hydrophilic groups in the PNIPAM macromolecular chain, and they will interact with water within and between molecules. When the temperature is lower than 32 °C, the phthalamide group interacts with water molecules via hydrogen bonds, so that the water molecules around PNIPAM form a solvation layer connected by hydrogen bonds. At this point, the macromolecular chain appears in an extended state. When the temperature is higher than 32 °C, the hydrogen bonds of the solvation layer break and the solvation layer is destroyed. At this point, the macromolecular chain is mutated from a stretched state to a tight state. Therefore, the volume of the PNIPAM hydrogel mutates near 32 °C, i.e., the lowest critical dissolution temperature (LCST) (Figure 3c(1)).

In contrast, the volume of PEGDA hydrogel cannot respond to the change in temperature significantly (Figure 3c(1)). The PNIPAM–PEGDA bilayer strip was fabricated via a secondary UV curing technique. No delamination occurred due to the covalent bond between the two layers. After experiments, we found that the PNIPAM–PEGDA bilayer structure has a bidirectional temperature response property. When temperature was below the LCST, the volume of PNIPAM layer expanded, and the volume of PEGDA layer remained almost completely unchanged; the bilayer structure bent, and the outer layer was PNIPAM layer while the inner layer was PEGDA layer. The volume of PNIPAM layer decreased with the increase in temperature, and the bending angle of the bilayer structure decreased. When the temperature reached the LCST, the bilayer strip fully extended. As the temperature continued to increase, the volume of PNIPAM layer continued to decrease. At this point, the outer layer was the PEGDA layer and the inner layer was the PNIPAM layer. We defined the outer PNIPAM layer as “positive” and the opposite as “negative”. As illuminated in Figure 3d(1), the bilayer structure bent regardless of whether the temperature was above or below the LCST, and the maximum value of both reached 180°.

In addition to the temperature, the PNIPAM layer can also respond to a change in solvent composition. In pure DI water or pure ethanol solution, the volume of PNIPAM hydrogel reached the maximum. In the mixed solution of ethanol and DI water, the volume of the PNIPAM hydrogel decreased. However, no matter the proportion of ethanol and DI water, the volume of PNIPAM hydrogel was less than that in pure DI water or pure ethanol (Figure 3c(2)). However, the volume of PEGDA hydrogel could not respond to the change in solvent composition. The PNIPAM–PEGDA bilayer structure was found to respond to different solvent compositions. In either ethanol or pure DI water, the bilayer structure was in a “positive” bending state. In the mixed solution, it presented a long strip (Figure 3d(2)).

In summary, since the PNIPAM layer is sensitive to both temperature and solvent composition, it is called the active layer. The PEGDA layer, on the other hand, is insensitive to both temperature and solvent composition and, therefore, is called the rigid layer.

### 3.4. Opening and Closing of Dual Responsive Bionic Flower

Based on the research on the deformation characteristics of the PNIPAM–PEGDA bilayer structure in Section 3.3, we fabricated a bionic flower with a bilayer structure. 

First, we designed the photomask of the bionic flower. As shown in Figure 4a(1), the bionic flower photomask contains 10 petals, and each petal has a black microchannel, forming the internal non-uniform stress field of the bionic flower conducive to the closure of the flower. We fabricated a single-layer PNIPAM bionic flower with a thickness of 400 μm using the photomask, and the single-layer bionic flower appeared white. It was placed in DI water at room temperature and initially floated on the surface of water due to light gravity. As the side on the water gradually absorbed water and expanded in volume, the volume of water on the side exposed to the air remained unchanged, and thus the single-layer bionic flower closed towards the side exposed to the air (Figure 4a(2)). However, the deformation was transient. When the absorbent volume of the on-water side expanded and gravity increased, the bionic flower sank into the water. At this point, the water absorption volume of the previously above-water side gradually expanded, and the deformation of the bionic flower recovered. To maintain the 3D structure of the bionic flower, we made bionic flowers based on the PNIPAM–PEGDA bilayer structure (Figure 4a(3)). Firstly, the thickness of the PI spacer shown in Figure 2b(1) was adjusted to 800 μm, and a PNIPAM bionic flower with a thickness of 400 μm was placed on a glass slide. Then, the PEGDA hydrogel prepolymer solution was dropped onto the surface of PNIPAM sheet. Then, the other glass slide was covered and cured by UV light. This process took about 10 s. The cured PEGDA layer was colorless and transparent. Subsequently, the bionic flower was given photothermal response characteristics by impregnating CNTs, as described in Section 3.1. The impregnated PNIPAM layer appeared dark gray and did not fade even after several deformation experiments, indicating that CNTs were firmly captured in the micropores of the PNIPAM. However, the impregnated PEGDA layer appeared light gray and the color faded after gentle wiping, indicating that there were fewer micropores in the PEGDA, and the CNTs could not be firmly captured. Due to the small number of micropores, there was almost no water entering or leaving from the inside of the PEGDA, which explains why the volume of the PEGDA is not sensitive to changes in temperature and solvent.

Due to the high swelling ratio of the active layer in pure water, the flower was normally closed in pure DI water at room temperature (about 25 °C). After injecting a certain amount of ethanol solution into the container, the swelling ratio of the active layer decreased, the volume decreased, and the bionic flower opened gradually. This process took about 70 s. The open flower was subsequently placed in pure DI water (about 25 °C); the active layer reabsorbed water and increased in volume, causing the bionic flower to reclose. This process took approximately 80 s (Figure 4b(1)). In addition, based on the temperature response characteristics of the bilayer structure, we achieved controllable opening and closing of bionic flower. As shown in Figure 4b(2), the active layer of the bionic flower was continuously illuminated with a laser light, the CNTs converted light into heat, the temperature increased and the volume of it decreased, and the bionic flower gradually opened. After removing the laser, the temperature decreased, the active layer gradually reabsorbed water and recovered to the original volume, and the bionic flower gradually closed again.

### 3.5. Intelligent Transportation Based on the Bionic Venus Flytrap Soft Microrobot

The Venus flytrap is a popular insectivorous plant, complete with root, stem, leaf, flower, and seed. Its leaves are the most prominent and obvious parts, which can hunt insects. The seta and red sessile glands look like bloody mouths (Figure 5a(1),(2)). Inspired by the Venus flytrap, we designed and fabricated a bilayer bionic Venus flytrap soft robot with several directional microchannels. The photomask of the soft robot is shown in Figure 5b, and there were several lateral microchannels on the two leaves and petiole to achieve a better closing movement. In addition, there were several spines on the edges of the two bionic leaves to make the soft robot more capable of wrapping. A PNIPAM layer with a thickness of 400 μm, i.e., the active layer, was made by UV-curing technology. The active layer was deformed via illumination and, after removing the laser, the deformation was recovered from. This maintained the 3D structure of the robot at all times, even in the absence of stimulations. Subsequently, the PEGDA layer, i.e., the rigid layer, was produced on the PNIPAM layer by secondary light curing. The robot was approximately 17 mm in length and 25 mm in width, and the diameter of the deformed tubular structure was approximately 6 mm. The soft robot achieved magnetic and photothermal conversion characteristics via dyeing in DI water dispersed with CNTs and magnetic Fe_3_O_4_ particles. The surface of the active layer appeared as a uniform black, indicating strong photothermal conversion ability. However, the surface of the rigid layer appeared light gray, indicating poor photothermal conversion ability. The explanation of the above phenomenon is that the PNIPAM hydrogel has larger and more abundant micropores on both its surface and interior than the PEGDA hydrogel, and thus more CNTs and Fe_3_O_4_ can be attached more firmly. The bionic Venus flytrap robot normally closed to become a tubular structure in cold water, with an outer active layer and an inner rigid layer (Figure 5c(1)–(3)). 

When the active layer of the bionic robot was continuously illuminated by a laser with an intensity of 1.2 W/cm^2^, the CNTs on the surface of the active layer converted light into heat, the volume decreased, and the soft robot gradually opened. After removing the laser, heat transferred to the surrounding water rapidly, and the robot closed again. Taking advantage of this property, we came up with the idea of using the excellent wrapping ability of the robot to achieve intelligent transportation of objects. As shown in Figure 5d, an object was wrapped in the normally closed tubular robot in cold water (T = 22 °C). By applying an external magnetic field, the robot moved to the target position. As a result of the existence of seta, the object was firmly packaged, ensuring that it could not fall off in the transport process. Subsequently, the active layer was continuously illuminated with a laser, the robot gradually opened, and the object was released from the robot (Appendix A).

In addition, we found that the robot was not only normally in a closed state in cold water, but also in hot water. This phenomenon can be explained by the research on the swelling ratio of PNIPAM and PEGDA hydrogel in Section 3.3. As shown in Figure 5e(1), in hot water (T = 37 °C), the robot deformed into a tubular structure. Driven by an external magnetic field, the robot achieved programmable movement (Appendix A). We recognized that the robot could be used for fixed-point transport drugs in the intestine and stomach. The average temperature of the human intestine and stomach is 37.5 to 38 °C, thus, in the intestine and stomach, the bionic flytrap robot would be in the normally closed state. The robot’s tubular cavity was used to hold drugs, and moved to the target position in the stomach model under a magnetic force, as depicted in Figure 5e(2). After arriving at the target position, the drugs gradually released under the effect of the gastric juice (Appendix A).

The experiments shown in Figure 5d,e were based on permanent magnets, which controlled the robot’s motion. The advantages of the permanent magnet are low cost and no power consumption, but there are disadvantages, such as low field strength and large weight. Subsequently, we drove the robot based on the magnetic field generated by the 3D Helmholtz coil. As shown in Figure 5f(1),g(1), the robot was placed, respectively, in a Petri dish and a stomach model with 37 °C hot water. Sinusoidal currents with a phase difference of 90° were input to two sets of coils, and the rotating magnetic field generated by the coils drove the rotation of the robot (Appendix A). In addition, the uniform magnetic field generated based on the Helmholtz coil drove the robot in a straight line (Figure 5f(2),g(2)). By controlling the phase difference, the movement direction of the robot could be changed. By varying the magnitude of the input current and the input voltage, the speed of rotation and straight forward motion could be controlled. The 3D Helmholtz coil has the advantages of simple operation, high precision, and good magnetic field stability.

Next, we discuss the recycling of the soft robot. The soft robot can deform reversibly. If the soft robot is used for intelligent transportation outside the body, it can be recycled repeatedly. If it is used for targeted drug delivery in vivo, its recycling has not been studied. Since the soft robot cannot be metabolized by the human body, we currently envision two schemes. The first one is to wrap the soft robot in a soluble capsule, swallow it, and control its movement to the target position via an external magnetic field for drug release. After releasing the drug completely, the soft robot enters the bowels and exits the body following intestinal peristalsis. Since the PNIPAM and PEGDA hydrogels are biocompatible, they are not harmful to the human body. However, this scheme has the disadvantage that the soft robot cannot be recycled and can only be used once. 

Another scheme is to use the soft robot for targeted drug delivery in blood vessels. This would mean reducing the size of robot to millimeters or smaller and injecting it into the blood vessel via a needle injection method. Thereafter, it follows an external magnetic field to reach the target position for drug delivery. After the completion of the targeted drug delivery, it is withdrawn with a needle tube.

At present, these two schemes have been conceived and not practiced in vivo. If one day they could be used in vivo, we do not think it would be necessary to consider recycling. On the one hand, the manufacturing cost of the soft robot is relatively low. On the other hand, considering the health issues, such as whether the patient is carrying certain infectious diseases, we do not recommend recycling the soft robot.

## 4. Conclusions

In this research, we studied the response characteristics of PNIPAM and PEGDA hydrogels. Since the PNIPAM hydrogel is temperature-sensitive and solvent-responsive, and the PEGDA hydrogel is insensitive to changes in both temperature and solvent composition, the bilayer structure can produce a bidirectional response to temperature and solvent composition. This can not only achieve a bidirectional transition of 2D-3D, but also maintain the 3D structure without continuous stimulations. Based on the bilayer structure, we fabricated a bionic Venus flytrap soft robot. The robot appeared as a normally closed tubular structure in either cold or hot water. With the addition of CNTs and Fe_3_O_4_, the robot possessed a photothermal conversion capability and magnetic response characteristic. The soft robot opened gradually when the active layer was illuminated, and the robot closed again after removing the laser. In addition, driven by an external magnetic field, the robot moved along a specific trajectory to the target position. We illustrated the intelligent transport capacity of the soft robot in both cold and hot water, and demonstrated the fixed-point transportation of the soft robot in the stomach model. We believe that this robot will promote the development of soft robots and provide a reference value for future targeted drug delivery technology in biomedicine.

## Figures and Tables

**Figure 1 biomimetics-08-00429-f001:**
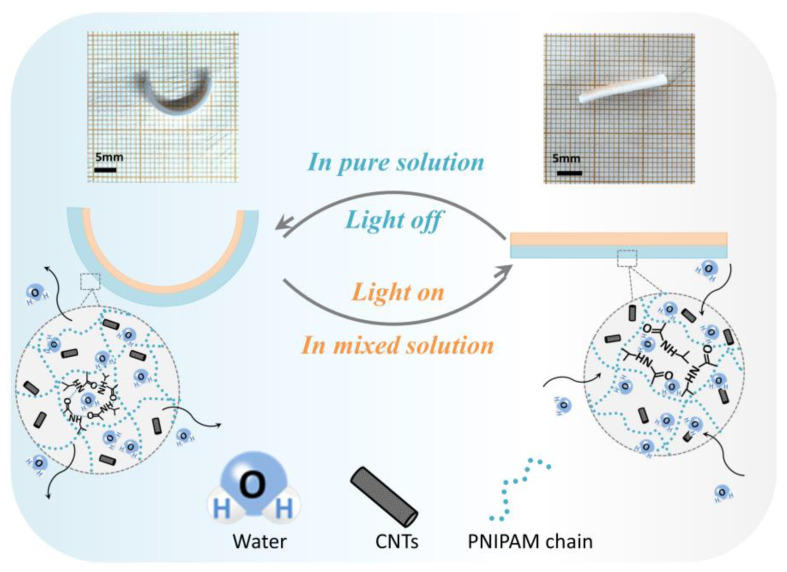
PNIPAM–PEGDA bilayer structure responding not only to a change in temperature, but also to a change in solvent composition (scale bar = 5 mm).

**Figure 2 biomimetics-08-00429-f002:**
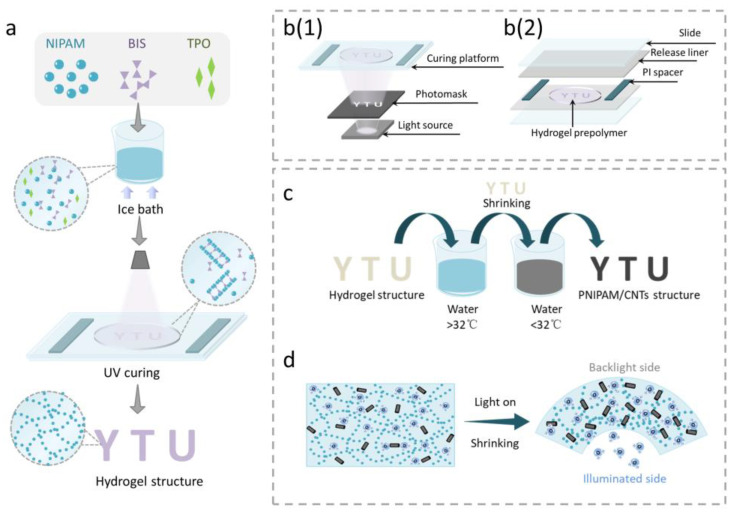
(**a**) Preparation of PNIPAM prepolymer. (**b**(**1**)) UV curing device. (**b**(**2**)) Structural composition of UV curing platform. (**c**) Impregnation of CNTs giving the hydrogel sheet photothermal characteristics. (**d**) Photothermal deformation of PNIPAM sheet impregnated with CNTs.

**Figure 3 biomimetics-08-00429-f003:**
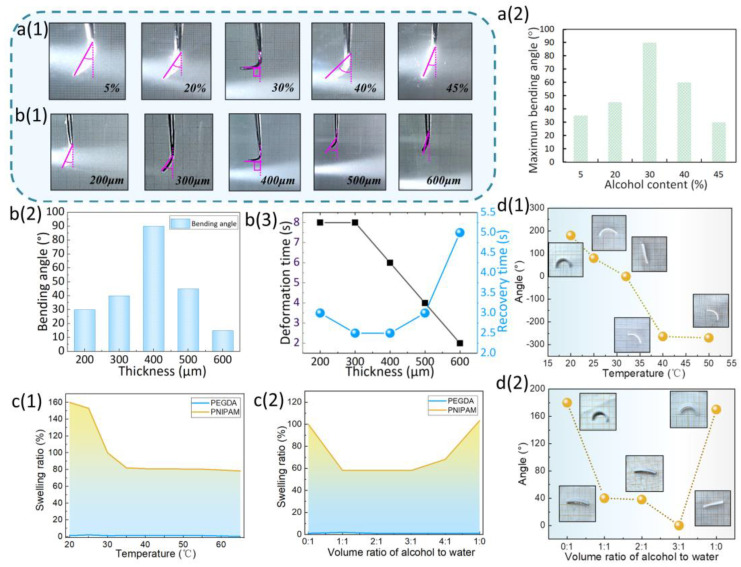
(**a**(**1**)) PNIPAM strips with different alcohol content responding to light. (**a**(**2**)) Relationship between alcohol content and the maximum bending angle of PNIPAM strip. (**b**(**1**)) PNIPAM strips with different thicknesses responding to light. (**b**(**2**)) Relationship between thickness and the maximum bending angle of PNIPAM strip. (**b**(**3**)) Relationship between thickness and the deformation/recovery time of PNIPAM strip. (**c**(**1**)) Relationship between temperature and the swelling ratio of PNIPAM/PEGDA hydrogels. (**c**(**2**)) Relationship between solvent composition and the swelling ratio of PNIPAM/PEGDA hydrogels. (**d**(**1**)) Relationship between temperature and the bending angle of PNIPAM–PEGDA bilayer hydrogel. (**d**(**2**)) Relationship between solvent composition and the bending angle of PNIPAM–PEGDA bilayer hydrogel.

**Figure 4 biomimetics-08-00429-f004:**
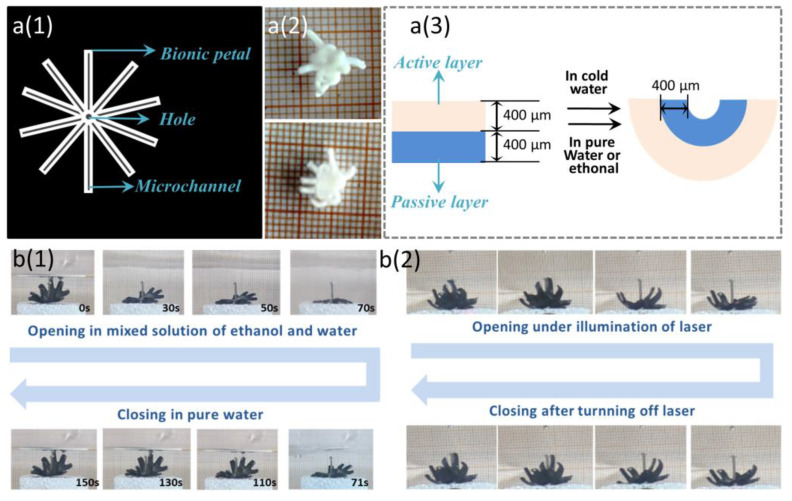
(**a**(**1**)) Photomask of bionic flower. (**a**(**2**)) Deformation of single-layer bionic flower. (**a**(**3**)) Structure diagram of the bilayer bionic flower. (**b**(**1**)) Bionic flower responding to the change in solvent composition. (**b**(**2**)) Bionic flower responding to the laser.

**Figure 5 biomimetics-08-00429-f005:**
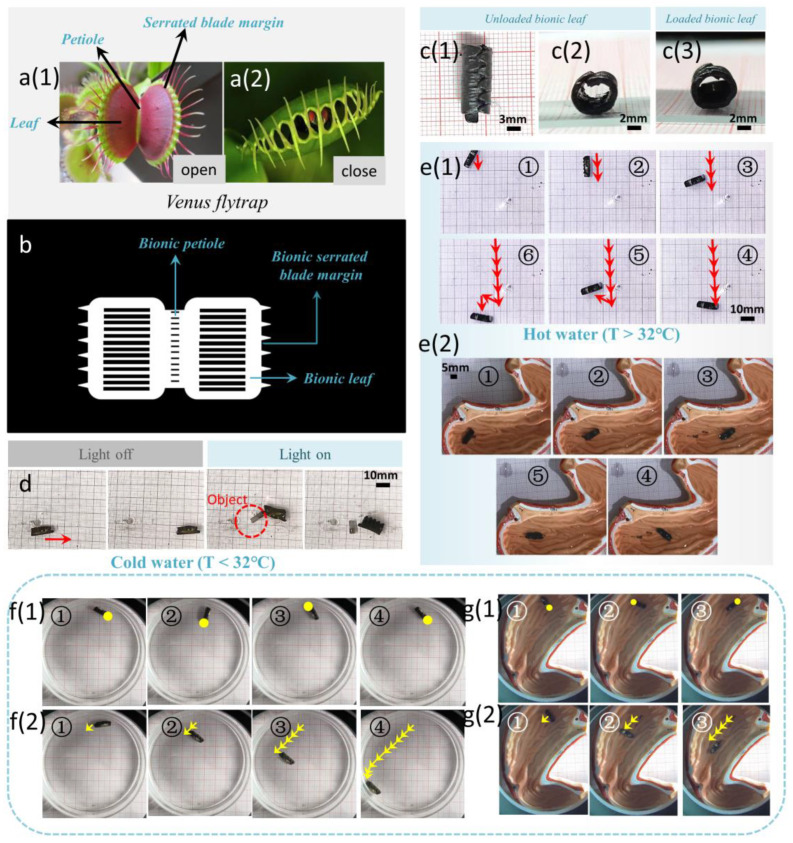
(**a**(**1**),**a**(**2**)) Structural composition of Venus flytrap. (**b**) Photomask of bionic Venus flytrap soft robot. (**c**(**1**),**c**(**2**)) Tube-shaped soft robot. (**c**(**3**)) Tube-shaped soft robot loading object. (**d**) Intelligent transportation of the soft robot in cold water. (**e**(**1**)) Intelligent transportation of the soft robot in hot water. (**e**(**2**)) Intelligent transportation of the soft robot in the stomach model. (**f**(**1**)) In hot water, the soft robot is driven to rotate by a magnetic field generated by a 3D Helmholtz coil. (**f**(**2**)) The soft robot is driven in a straight line by a magnetic field generated by a 3D Helmholtz coil. (**g**(**1**)) In a stomach model equipped with hot water, the soft robot is driven to rotate by a magnetic field generated by a 3D Helmholtz coil. (**g**(**2**)) In a stomach model equipped with hot water, the soft robot is driven in a straight line by a magnetic field generated by a 3D Helmholtz coil.

## Data Availability

Data available on request from the authors.

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
