# Peer review of "A Bionic Venus Flytrap Soft Microrobot Driven by Multiphysics for Intelligent Transportation"

_biomimetics, 2023, doi:10.3390/biomimetics8050429_

Round 1
Reviewer 1 Report
In my opinion, the manuscript is not suitable for publication. I started my evaluation by watching the attached videos. The first thing that catches the eye is that the position control of the microrobot using a magnetic field is not fully explained - probably the source of this field is very close, hidden behind a card with a millimeter scale - it's hard not to associate it with a commonly known toy magnetic activity. As for the provided video with a model of the stomach: in a real situation, access of a localized magnetic field to a given organ requires its special spatial shape - the authors did not focus on this. Moreover, in the article I did not find a presentation of really new materials. The presented polymeric materials and their behavior have been known for many years; I found an examplenary publication from 2017 very quickly. Also, the idea of transporting microcapsules to a given place in the human body, using a localized magnetic field, is not something original today. Thus, in its current form, the article is not suitable for publication.
Author Response
Dear Editor and Reviewers:
Thank you for your letter and for the reviewer’s comments concerning our manuscript. Those comments are all valuable and very helpful for revising and improving our paper, as well as the important guiding significance to our researches. We have studied the comments carefully and have made correction which we hope meet with approval. Revised portion are marked in red in the paper. The main corrections in the paper and the responds to the reviewer’s comments are as follows:
Responds to the reviewer’s comments:
Reviewer 1#:
Review report:
Comments to the Author
- In my opinion, the manuscript is not suitable for publication. I started my evaluation by watching the attached videos. The first thing that catches the eye is that the position control of the microrobot using a magnetic field is not fully explained - probably the source of this field is very close, hidden behind a card with a millimeter scale - it's hard not to associate it with a commonly known toy magnetic activity.
Response: We thank the reviewer for the comment.
The focus of this study is to realize intelligent transportation based on the temperature responsiveness of the soft robot, so we only demonstrated through simple experiments that the motion trajectory of the soft robot can be controlled by magnetic drive. In this experiment, we originally only used a permanent magnet to generate a magnetic field (magnetic field strength less than 1T) to drive the movement of robot. But the movement form of robot based on the permanent magnet was relatively single.
According to reviewer’s suggestion, we have driven the robot based on a 3D Helmholtz coil to achieve rotational and linear motion. Helmholtz coil-based actuation is easy to operate, highly controllable, and magnetic field stable. We have marked it in red in the paper.
3.5
“The experiments shown in Figure d and e were based on permanent magnets to drive the robot motion. The advantages of permanent magnet are low cost and no power consumption, but there are disadvantages such as low field strength and large weight. Subsequently, we drove the robot based on the magnetic field generated by the 3D Helmholtz coil. As shown in Figure 5f1 and g1, the robot was respectively placed in a Petri dish and a stomach model with 37℃ hot water. Sinusoidal currents with a phase difference of 90° were input to two sets of coils, and the rotating magnetic field generated by the coils drove the rotation of the robot. In addition, the uniform magnetic field generated based on the Helmholtz coil drove the robot in a straight line (Figure 5f2 and g2). By controlling the phase difference, the movement direction of the robot could be changed. By varying the magnitude of the input current and the input voltage, the speed of rotation and straight forward could be controlled. 3D Helmholtz coil has the advantages of simple operation, high precision and good magnetic field stability.”
- As for the provided video with a model of the stomach: in a real situation, access of a localized magnetic field to a given organ requires its special spatial shape - the authors did not focus on this.
Response: We thank the reviewer for the comment.
In this study, we propose a bionic Venus flytrap soft robot with bilayer structure for intelligent transportation. The robot is temperature responsive and magnetic responsive, which can convert from a 2D sheet to a 3D tubular structure reversibly. It is normally closed in both cold (T<32℃) and hot water (T>32℃) and can be used to load and transport objects to the target location (magnetic field strength<1T). Furthermore, based on the performance of the soft robot, we demonstrated the site-directed transportation of the robot in a gastric model. This study provides a reference for the robot for targeted drug release, and we expect that the robot can be improved to achieve targeted drug release in the human body in the future. Therefore, this research is currently in the theoretical research stage, and the problem of the spatial shape of the local magnetic field in the real situation has not been considered.
- Moreover, in the article I did not find a presentation of really new materials. The presented polymeric materials and their behavior have been known for many years; I found an examplenary publication from 2017 very quickly.
Response: We thank the reviewer for the comment.
The materials used in this study were PNIPAM hydrogel and PEGDA hydrogel. Although both PNIPAM and PEGDA hydrogels are not the latest materials, they are still popular among researchers due to their excellent biocompatibility, reversibility, and water-retaining property. There are several researches based on these two hydrogels carried out by researchers of all over the world, promoting the development of biomedicine and intelligent robotics.
In this study, the temperature sensitive characteristic of PNIPAM hydrogel and the temperature insensitive characteristic of PEGDA were combined to drive the deformation of soft robot to achieve intelligent transportation. PNIPAM is a kind of smart material for a wide range of applications due to its fast response properties and biocompatibility. Therefore, we chose these two kinds of materials as raw materials for the soft robot in this study.
- Also, the idea of transporting microcapsules to a given place in the human body, using a localized magnetic field, is not something original today. Thus, in its current form, the article is not suitable for publication.
Response: We thank the reviewer for the comment.
Although the idea of transporting microcapsules to a given place in the human body, using a localized magnetic field has been proposed before, it has not been widely used in clinical practice so far. Therefore, we believe that this field should be explored continuously. In this study, we fabricated a soft robot based on a bilayer structure, which was normally closed in either cold or hot water. The soft robot based on hydrogel is low cost and harmless to the human body, and the soft structure avoids the harm of rigid equipment to the human body. We expect it to be used for targeted drug delivery in the future, and we are not aware of soft robot with similar construction made previously.
Once again, thank you very much for your comments and suggestions.

Reviewer 2 Report
The article by Wang et al. investigated a hydrogel bilayer structure. The soft robot based on the bilayer structure had good stretching and directional motion properties under the action of multiple physical fields. And the performance tests reflected the characteristics of the robot well. I would like to suggest publication after carefully addressing the following comments.
1. The Abstract section did not highlight the focus of the research. It should include the structure and properties of the robot, as well as the specific physical fields (magnetic, temperature, concentration of the solution). Besides, the first two sentences were redundant and could be placed in the Introduction section.
2. The authors should improve the clarity of Figure 1, the chemical structure formulas of the two figures below were too small to read.
3. On page 4, line 122, “After turning off the light, the heat rapidly transferred to the surrounding water, allowing the illuminated side to absorb water again and the deformation recovered.” The causality of this statement was not clear enough. Heat dissipation made the temperature drop, causing the temperature-responsive hydrogel to swell and thus absorbed water and recovered.
4. There were a few beams of light in Figures 3a1 and 3b1 that did not shine on the sheet, was this still a contrasting effect? In addition, it is necessary to indicate the datum line for the angular measurement. For example, in 3a1 the start and end points of the angle measurement were indicated by specific lines. Some of the diagrams need to be made clearer and more contrasting, such as those corresponding to 40 ℃ in Figure 3d1, and 3:1 and 1:1 in 3d2.
5. The authors should add a diagram of the bionic flower, including the materials corresponding to the upper and lower layers. The text description section needs to be more detailed (colors and materials for each layer), like the description on page 7, line 238.
6. Figure 5e2 did not indicate the order.
7. How was this drug carrier recycled and was it only for single use? And how biocompatible was this material and can it be metabolized by the body?
Please see above
Author Response
Dear Editor and Reviewers:
Thank you for your letter and for the reviewer’s comments concerning our manuscript. Those comments are all valuable and very helpful for revising and improving our paper, as well as the important guiding significance to our researches. We have studied the comments carefully and have made correction which we hope meet with approval. Revised portion are marked in red in the paper. The main corrections in the paper and the responds to the reviewer’s comments are as follows:
Responds to the reviewer’s comments:
Reviewer 2#:
Review report:
Comments to the Author
The article by Wang et al. investigated a hydrogel bilayer structure. The soft robot based on the bilayer structure had good stretching and directional motion properties under the action of multiple physical fields. And the performance tests reflected the characteristics of the robot well. I would like to suggest publication after carefully addressing the following comments.
- The Abstract section did not highlight the focus of the research. It should include the structure and properties of the robot, as well as the specific physical fields (magnetic, temperature, concentration of the solution). Besides, the first two sentences were redundant and could be placed in the Introduction section.
Response: We thank the reviewer for the comment.
According to reviewer’s suggestion, we have revised the abstract section to add the structure and properties of the robot, as well as the specific physical fields (magnetic field, temperature, solution concentration). In addition, we removed the first two sentences. We have marked it in red in the paper.
Abstract
“With the continuous integration of material science and bionic technology, as well as people's increasing requirements for the operation of robots in complex environments, researchers continue to develop bionic intelligent microrobots, whose development will bring great revolution to people's production and life. In this study, we propose a bionic flower based on the PNIPAM-PEGDA bilayer structure. PNIPAM is temperature responsive and solvent responsive, thus acts as an active layer, while PEGDA does not change significantly in response to the change of temperature and solvent, thus acts as a rigid layer. The bilayer flower is closed in the cold water and gradually opens under laser illumination. In addition, the flower gradually opens by injecting ethanol into water. When the volume of ethanol exceeds the volume of water, the flower opens completely. In addition, we propose a bionic Venus flytrap soft microrobot with a bilayer structure. The robot is temperature responsive and can reversibly transform from a 2D sheet to a 3D tubular structure. It is in the normally closed state in both cold (T<32℃) and hot water (T>32℃), which can be used to load and transport objects to the target position (magnetic field strength<1T).”
- The authors should improve the clarity of Figure 1, the chemical structure formulas of the two figures below were too small to read.
Response: We thank the reviewer for the comment.
The chemical structure formulas in Figure 1 are too small to see clearly.
According to reviewer’s suggestion, we have modified the figure and adjusted the size of the structure.
- On page 4, line 122, “After turning off the light, the heat rapidly transferred to the surrounding water, allowing the illuminated side to absorb water again and the deformation recovered.” The causality of this statement was not clear enough. Heat dissipation made the temperature drop, causing the temperature-responsive hydrogel to swell and thus absorbed water and recovered.
Response: We thank the reviewer for the comment. The explanation of causality between the light turning off and deformation recovery is not enough.
According to reviewer’s suggestion, we have detailed the causality between the light turning off and deformation recovery. We have marked it in red in the paper.
3.2
“When one side of the PNIPAM sheet was illuminated by light, the CNTs on the surface rapidly absorbed the light energy and converted it into heat, the illuminated side lost water and decreased in volume, while the volume of the backlit side remained unchanged, bending the hydrogel toward the light source. After turning off the light, the heat rapidly transferred to the surrounding water and the temperature dropped, causing the illuminated side of temperature responsive hydrogel to reabsorb water and the deformation recovered.”
- There were a few beams of light in Figures 3a1 and 3b1 that did not shine on the sheet, was this still a contrasting effect? In addition, it is necessary to indicate the datum line for the angular measurement. For example, in 3a1 the start and end points of the angle measurement were indicated by specific lines. Some of the diagrams need to be made clearer and more contrasting, such as those corresponding to 40 ℃ in Figure 3d1, and 3:1 and 1:1 in 3d2.
Response: We thank the reviewer for the comment.
In Figure 3a and b, the hydrogel strip was illuminated by light and the light was removed when the deformation reached the maximum. The strips without illumination shows the state when the deformation reached the maximum and the illumination was immediately removed, which was still the maximum deformation Angle.
According to reviewer’s suggestion, we have added both the datum line to both Figure 3a1 and b1, and increased the clarity of Figure 3d1 and d2.
- The authors should add a diagram of the bionic flower, including the materials corresponding to the upper and lower layers. The text description section needs to be more detailed (colors and materials for each layer), like the description on page 7, line 238.
Response: We thank the reviewer for the comment.
According to reviewer’s suggestion, we have added a diagram of the bionic flower, including the materials corresponding to the upper and lower layers. In addition, we have descripted the bilayer structure (colors and materials for each layer) in more detail. We have marked it in red in the paper.
3.4
“Based on the research of the deformation characteristics of PNIPAM-PEGDA bilayer structure in Section 3.3, we fabricated bionic flower with bilayer structure.
First, we designed the photomask of the bionic flower. As shown in Figure a1, the bionic flower photomask contains 10 petals, and each petal has a black microchannel, forming the internal non-uniform stress field of the bionic flower to conducive to the closure of the flower. We fabricated a single-layer PNIPAM bionic flower with a thickness of 400 μm by using the photomask, and the single-layer bionic flower appeared white. It was placed in DI water at room temperature and initially floated on the surface of water due to the light gravity. As the near water side gradually absorbed water and expanded in volume, the volume of water on the back water side remained unchanged, the single-layer bionic flower closed towards the back water side (Figure a2). However, the deformation was transient. When the absorbent volume of the near water side expanded and gravity increased, the bionic flower sank into the water. At this point, the water absorption volume of the back water side gradually expanded, and the deformation of the bionic flower recovered. To maintain the 3D structure of the bionic flower, we made bionic flowers based on PNIPAM-PEGDA bilayer structure (Figure a3). Firstly, the thickness of the PI spacer shown in Figure 2b2 was adjusted to 800 μm, and the PNIPAM bionic flower with a thickness of 400 μm was placed on the glass slide, then the PEGDA hydrogel prepolymer solution was dropped on the surface of PNIPAM sheet. Then the other glass slide was covered and cured by UV light. This process took about 10 seconds. The cured PEGDA layer was colorless and transparent. Subsequently, the bionic flower was given photothermal response characteristics by impregnating CNTs as described in Section 3.1. The impregnated PNIPAM layer appeared dark gray and did not fade even after several deformation experiments, indicating that CNTs were firmly captured in the micropores of PNIPAM. However, the impregnated PEGDA layer appeared light gray, and the color faded after gentle wiping, indicating that there were fewer micropores in PEGDA, and CNTs couldn’t be firmly captured. Due to the small number of micropores, there is almost no water entering or losing from the inside of PEGDA, which also explains why the volume of PEGDA is not sensitive to the change of temperature and solvent.
Due to the high swelling ratio of the active layer in pure water, the flower was normally closed in pure DI water at room temperature (about 25℃). After injecting a certain amount of ethanol solution into the container, the swelling ratio of active layer decreased and the volume decreased, the bionic flower opened gradually, and this process took about 70 seconds. The open flower was subsequently placed in pure DI water (about 25℃), the active layer reabsorbed water and increased in volume, causing the bionic flower to reclose, this process took approximately 80 seconds (Figure 4b1). In addition, based on the temperature response characteristics of the bilayer structure, we achieved the controllable opening and closing of bionic flower. As shown in Figure 4b2, the active layer of the bionic flower was continuously illuminated with a laser light, the CNTs converted light into heat, the temperature increased and the volume of it decreased, the bionic flower gradually opened. After removing the laser, the temperature decreased and active layer gradually reabsorbed water and recovered to the original volume, the bionic flower gradually closed again.”
- Figure 5e2 did not indicate the order.
Response: We thank the reviewer for the comment.
According to reviewer’s suggestion, we have indicated the order of Figure 5e2.
- How was this drug carrier recycled and was it only for single use? And how biocompatible was this material and can it be metabolized by the body?
Response: We thank the reviewer for the comment.
The soft robot can deform reversibly. If the soft robot is used for intelligent transportation outside the body, it can be recycled repeatedly. If it is used for targeted drug delivery in vivo, the recycling of it has not been studied.
Since the soft robot cannot be metabolized by human body, we currently envision two schemes. The first one is to wrap the soft robot in a soluble capsule, swallow it, and control its movement to the target position by an external magnetic field for drug release. After releasing the drug completely, the soft robot enters the enteropathy and exits the body following intestinal peristalsis. Since the PNIPAM and PEGDA hydrogels are biocompatible, they are not harmful to the human body. However, this scheme has the disadvantage that it cannot be recycled and can only be used once.
Another scheme is to use the soft robot for targeted drug delivery in blood vessels. Reducing the size of robot to millimeters or smaller, and injecting it into the blood vessel by needle injection method. Thereafter it follows by an external magnetic field to reach the target position for drug delivery. After the completion of the targeted drug delivery, it is withdrawn with a needle tube.
At present, these two schemes have been conceived and not practiced in vivo. If one day it could be used in vivo, we don't think it would be necessary to consider recycling. On the one hand, the manufacturing cost of the soft robot is relatively low. On the other hand, considering the health issues, such as whether the patient is carrying certain infectious diseases, we do not recommend recycling the soft robot.
According to reviewer’s suggestion, we have explained the problem of the robot recycling detailedly. We have marked it in red in the paper.
3.5
“The experiments shown in Figure d and e were based on permanent magnets to drive the robot motion. The advantages of permanent magnet are low cost and no power consumption, but there are disadvantages such as low field strength and large weight. Subsequently, we drove the robot based on the magnetic field generated by the 3D Helmholtz coil. As shown in Figure 5f1 and g1, the robot was respectively placed in a Petri dish and a stomach model with 37℃ hot water. Sinusoidal currents with a phase difference of 90° were input to two sets of coils, and the rotating magnetic field generated by the coils drove the rotation of the robot. In addition, the uniform magnetic field generated based on the Helmholtz coil drove the robot in a straight line (Figure 5f2 and g2). By controlling the phase difference, the movement direction of the robot could be changed. By varying the magnitude of the input current and the input voltage, the speed of rotation and straight forward could be controlled. 3D Helmholtz coil has the advantages of simple operation, high precision and good magnetic field stability.
Next, we discuss the recycling of the soft robot. The soft robot can deform reversibly. If the soft robot is used for intelligent transportation outside the body, it can be recycled repeatedly. If it is used for targeted drug delivery in vivo, the recycling of it has not been studied. Since the soft robot cannot be metabolized by human body, we currently envision two schemes. The first one is to wrap the soft robot in a soluble capsule, swallow it, and control its movement to the target position by an external magnetic field for drug release. After releasing the drug completely, the soft robot enters the enteropathy and exits the body following intestinal peristalsis. Since the PNIPAM and PEGDA hydrogels are biocompatible, they are not harmful to the human body. However, this scheme has the disadvantage that it cannot be recycled and can only be used once.
Another scheme is to use the soft robot for targeted drug delivery in blood vessels. Reducing the size of robot to millimeters or smaller, and injecting it into the blood vessel by needle injection method. Thereafter it follows by an external magnetic field to reach the target position for drug delivery. After the completion of the targeted drug delivery, it is withdrawn with a needle tube.
At present, these two schemes have been conceived and not practiced in vivo. If one day it could be used in vivo, we don't think it would be necessary to consider recycling. On the one hand, the manufacturing cost of the soft robot is relatively low. On the other hand, considering the health issues, such as whether the patient is carrying certain infectious diseases, we do not recommend recycling the soft robot.”
Once again, thank you very much for your comments and suggestions.

Reviewer 3 Report
This manuscript proposes a bionic Venus flytrap soft microrobot with a bilayer structure. The robot is temperature-responsive and can reversibly transform from a 2D sheet to a 3D tubular structure that can be used to load and transport objects to the target position under the drive of a magnetic field. This paper is detailed and interesting. Before being published, authors should address the following questions.
1. The authors are advised to add more references published in recent years in the introduction to support background.
2. The text in the figures is too small and the figures are not clear. The authors are advised to revise figures.
3. The dimensions of the robot is missing? This data is a major influence relating to the design and applications.
4. The Intelligent transportation based on the bionic Venus flytrap soft microrobot were achieved in Figure 5, the experimental temperatures should be clearly stated.
5. The grammar of the paper should be checked and corrected.
The grammar of the paper should be checked and corrected.
Author Response
Dear Editor and Reviewers:
Thank you for your letter and for the reviewer’s comments concerning our manuscript. Those comments are all valuable and very helpful for revising and improving our paper, as well as the important guiding significance to our researches. We have studied the comments carefully and have made correction which we hope meet with approval. Revised portion are marked in red in the paper. The main corrections in the paper and the responds to the reviewer’s comments are as follows:
Responds to the reviewer’s comments:
Reviewer 3#:
Review report:
Comments to the Author
This manuscript proposes a bionic Venus flytrap soft microrobot with a bilayer structure. The robot is temperature-responsive and can reversibly transform from a 2D sheet to a 3D tubular structure that can be used to load and transport objects to the target position under the drive of a magnetic field. This paper is detailed and interesting. Before being published, authors should address the following questions.
- The authors are advised to add more references published in recent years in the introduction to support background.
Response: We thank the reviewer for the comment.
According to reviewer’s suggestion, we have added more references published in recent years in the introduction to support background.
- The text in the figures is too small and the figures are not clear. The authors are advised to revise figures.
Response: We thank the reviewer for the comment.
According to reviewer’s suggestion, we have revised these figures to make them clearer.
- The dimensions of the robot is missing? This data is a major influence relating to the design and applications.
Response: We thank the reviewer for the comment.
According to reviewer’s suggestion, we added the dimensions of the soft robot. We have marked it in red in the paper.
3.5
“Subsequently, PEGDA layer, i.e. rigid layer, was produced on the PNIPAM layer by secondary light curing. The robot was approximately 17mm in length and 25mm in width, and the diameter of the deformed tubular structure was approximately 6mm. The soft robot has magnetic and photothermal conversion characteristics by dyeing in DI water dispersed with CNTs and magnetic Fe3O4 particles.”
- The Intelligent transportation based on the bionic Venus flytrap soft microrobot were achieved in Figure 5, the experimental temperatures should be clearly stated.
Response: We thank the reviewer for the comment.
According to reviewer’s suggestion, we have stated the experimental temperatures. We have marked it in red in the paper.
3.5
“When the active layer of the bionic robot was continuously illuminated by a laser with an intensity of 1.2 W/cm2, the CNTs on the surface of active layer converted light into heat, the volume decreased, and the soft robot gradually opened. After removing the laser, heat transferred to the surrounding water rapidly, and the robot closed again. Taking advantage of this property, we came up with the idea of using the excellent wrapping ability of the robot to achieve intelligent transportation of objects. As shown in Figure 5d, there was an object wrapped in the normally closed tubular robot in cold water (T=22℃). By applying an external magnetic field, the robot exercised to the target position. As a result of the existence of seta, the object was firmly packages, ensuring that couldn’t fall off in the transport process. Subsequently, the active layer was continuously illuminated with a laser, the robot gradually opened and the object was released from the robot (Video S1).
In addition, we found that the robot was not only in a normally closed state in the cold water, but also in hot water. This phenomenon can be explained by the research of the swelling ratio of PNIPAM and PEGDA hydrogel in Section 3.3. As shown in Figure 5e1, in hot water (T=37℃), the robot deformed into tubular structure.”
- The grammar of the paper should be checked and corrected.
Response: We thank the reviewer for the comment.
According to reviewer’s suggestion, we have checked and corrected the grammar of the paper. We have marked it in red in the paper.
Once again, thank you very much for your comments and suggestions.

Round 2
Reviewer 1 Report
The authors of the manuscript convinced me of the importance of the content provided. I think that the published article will be a positive source of inspiration for the scientific world for further research in this area.